# Lifestyle in Obese Individuals during the COVID-19 Pandemic

**DOI:** 10.3390/healthcare10091807

**Published:** 2022-09-19

**Authors:** Giovanna Pavone, Nicola Tartaglia, Michele De Fazio, Vincenzo Monda, Anna Valenzano, Giuseppe Cibelli, Rita Polito, Maria Ester La Torre, Fabio Scattarella, Laura Mosca, Alessia Scarinci, Gennaro Martines, Mario Pacilli, Giovanni Messina, Marcellino Monda, Antonietta Messina, Antonio Ambrosi

**Affiliations:** 1Department of Medical and Surgical Sciences, University of Foggia, Viale Pinto, 71122 Foggia, Italy; 2Department of General Surgery and Liver Transplantation “M. Rubino”, University of Bari, 70124 Bari, Italy; 3Department of Experimental Medicine, Section of Human Physiology and Unit of Dietetics and Sports Medicine, Università degli Studi della Campania “Luigi Vanvitelli”, 80138 Naples, Italy; 4Department of Clinical and Experimental Medicine, University of Foggia, Viale Pinto, 71100 Foggia, Italy; 5Department of Advanced Medical and Surgical Sciences, Università degli Studi della Campania “Luigi Vanvitelli”, 80138 Naples, Italy; 6Department of Education Sciences, Psychology, Communication, University of Bari, 70121 Bari, Italy

**Keywords:** SARS-CoV-2, obesity, mental health, lock-down, bariatric surgery, pandemic

## Abstract

Background: Stay-at-home orders in response to the Coronavirus 2 (SARS-CoV-2) pandemic have forced abrupt changes to daily routines. The aim of this study is to describe the behavior of lifestyles of individuals with obesity on the waiting list for bariatric surgery in the Department of Medical and Surgical Sciences of University of Foggia during the COVID-19 pandemic. Materials and methods: From June 2020 to December 2020 an online survey format was administered to all the patients (n = 52) enrolled for bariatric surgery subjects with obesity, to obtain information about the COVID-19 pandemic’s impact on patients with obesity starting 9 March 2020 until 18 May 2020. Results: Our data showed that 58% of patients stated that the pandemic negatively affected their mood, 60% of patients confirmed that they changed their dietary behaviors during the stay-at-home period, as they consumed more unhealthy foods or spent less time cooking home cooked meals. In addition, 71% of patients stated that the closure of the gyms worsened their obesity condition and their mental well-being with an increase of a feeling of anxiety. Conclusions: Results showed that the COVID-19 pandemic has had a significant impact on health behaviors, including quality of life, mental health physical activity, weight maintenance, and consumption of sweets in obese patients.

## 1. Introduction

The severe acute respiratory syndrome coronavirus 2 (SARS-CoV-2) is caused by the Coronavirus-19 disease (COVID-19) [1,2,3,4]. According to the World Health Organization (WHO), this virus has been declared a pandemic and it has a strong impact on lifestyle [5,6].

Several recent studies have reported that COVID-19 caused the destruction of lung parenchyma, including interstitial inflammation and extensive consolidation [7,8,9,10,11].

In addition, the host predisposition represents an important factor to prevent and/or treat this disease. Obesity is a multifactorial disease characterized by energy imbalance. Obesity is linked to an accumulation of visceral adipose tissue that leads to systemic chronic low-grade inflammation [12]. For this reason, obesity is strongly associated to metabolic and inflammatory pathologies, such as cardiovascular diseases, metabolic syndrome, immune disorders, and also cancer development [13]. Obese patients are more exposed to the risk of developing the most severe COVID-19 clinical features. Indeed, literature reports that severe obesity (i.e., BMI ≥ 40 kg/m^2^) is a common clinical risk factor for a worse prognosis and a higher mortality in patients with coronavirus disease 2019 (COVID-19) infection. Furthermore, any degree of obesity (BMI ≥ 30 kg/m^2^) has been associated with poor prognosis in patients with COVID-19 [14]. Obesity is a risk factor for increased severity and worse prognosis in patients with COVID-19 infection. Similarly, obesity-induced fatty tissue inflammation and its effects on the immune system play a crucial role in the pathogenesis of COVID-19 infection. In addition, it also causes metabolic dysfunctions, which can lead to dyslipidemia, insulin resistance, and cardiovascular diseases [15].

However, what is not known is how the COVID-19 “stay orders” such as self-quarantine, lockdown, and/or mandatory stay-at-home orders impacted mental and financial health, in addition to physical health of individuals with obesity [15,16]. Data literature reported that a strong association has been demonstrated between Nonalcoholic Fatty Liver Disease (NAFLD), obesity, and a general worsening of neuropsychological signs and symptoms [17].

The stay-at-home orders forced the cancellation of elective surgeries, including metabolic and bariatric surgery (MBS), and to date, the impact on this patient populations is not known [18].

Stay-at-home orders have curbed the spread of the virus, [19] yet the results of these unprecedented government mandates on other indices of health cannot be overlooked. The closure of workplaces, restaurants, fitness facilities, and other public places during the lockdown period imposed abrupt changes in the habits of the entire population. Furthermore, social isolation has had a detrimental impact on mental wellness [20]. Stress is associated with sleep disruption, consumption of highly palatable foods, and increased snacking, often resulting in weight gain [21]. The early collected data allowed evaluating the impacts of weight gain during the pandemic [22,23,24]. The lockdown imposed a change in people’s lives, lifestyles were modified, and above all, cases of obesity and sedentary lifestyle due to the restrictions of the lockdown increased.

In addition, higher scores for general behavioral deterioration, anxiety, delusions, hallucinations and apathy have been found, the direct consequence of which was a drastic increase in the number of psychiatric drugs prescribed and used during the lockdown [25].

This had effects not only on the incidence of new pathologies but also mental implications with an increase in pathologies related to the nervous system such as depression. Moreover, the obese subjects, from a psychological point of view, have a habit of not accept their own bodies and a loop is created, in which the more they have mental problems, the more they eat, aggravating the state of obesity even more in these subjects, who are waiting to undergo a bariatric surgery [13,14].

In the light of this evidence, the aim of this study is to describe the behavior of lifestyles of individuals with obesity on the waiting list for bariatric surgery in the Department of Medical and Surgical Sciences of University of Foggia during the COVID-19 pandemic. 

## 2. Materials and Methods

### 2.1. Participants

For this study, we enrolled 52 patients enlisted for bariatric surgery at the Department of Medical and Surgical Sciences of University of Foggia after the completion of a brief survey (87% of the total patients on the waiting list, the remaining 13% did not want to answer the questionnaire). 

Thirty-three percent of the patients were aged between 31 and 40 years, 25% between 21 and 30, 25% between 41 and 50 years, and 17% between 51 and 60 years. In addition, the majority (71%) of the patients were female and 29% were male. Thirty-nine percent of the patients were unemployed, 25% were employed, 20% were housewives, and 16% were students. Our Health System Institutional Review Board approved the survey. Those that agreed to participate signed an online consent form and authorized researchers to contact them for follow-up information. The UIN for ClinicalTrial.gov (accessed on 1 March 2022) Protocol Registration and Results System is: NCT05137405 for the Organization UFoggia. 

### 2.2. Eligibility Criteria

Adults of both genders with morbid obesity defined as BMI > 40 kg/m^2^ or BMI > 35 kg/m^2^ with at least one associated major comorbidity were included, according to the SICOB guidelines [26]. 

### 2.3. Study Protocol and Data Collection

From June 2020 to December 2020, an anonymous online survey format was administered to all individuals with obesity on the waiting list for bariatric surgery at the Department of Medical and Surgical Sciences of University of Foggia to obtain information about the COVID-19 pandemic’s impact on patients with obesity starting 9 March 2020 until 18 May 2020.

Obese patients were asked to answer a questionnaire with 16 closed questions about their experiences during the lockdown period, concerning their health and lifestyle behaviors. The questions were formulated in a clear and simple way so that they could be easily understood by all socio-cultural levels. The choice of online administration was driven by the pandemic crisis caused by SARS-CoV-2 and the logistical challenges of sending the card. The questions were formulated and proposed by all members of the team including the surgeon, the nurse, and the anesthetist. The questionnaire was carried out anonymously by the various participants in the study. 

### 2.4. Statistical Analysis

The domains of the survey were presented as demographics, sedentary behaviors, physical activity, diet, mental health. Descriptive statistics (means, frequencies, and percentage of the sample) were used to analyze all responses and are presented as frequency (percentage). The data were analyzed with Chi-square test with a *p* value two-tailed less than 0.05 (*p* < 0.05) for statistical significance.

## 3. Results

The COVID-19 pandemic affected the lifestyle of individuals answering the questionnaire. As shown in the Table 1, there is a statistical significance for the answer numbers 5, 6, 8, 12, 13, 15, and16, corresponding to lifestyle questions.

In particular, 58% of the patients stated that the pandemic negatively affected their mood, and 60% of them confirmed that they changed their dietary behaviors during the stay-at-home period, as they consumed more unhealthy food or spent more time cooking. Half of the obese people said they spent more time in the kitchen cooking (Table 1).

Seventy-one percent of individuals stated that the closure of the gyms influenced the worsening of their obesity condition and their mental well-being with an increase in the state of anxiety; in this case, it can be suggested that telematic support from a psychologist and a nutritionist would have been helpful.

Only 6% of obese individuals on the waiting list contracted the SARS-CoV-2 virus and none of those required hospitalization. 

During the lock-down period, all the participants were still convinced that they wanted to undergo bariatric surgery.

## 4. Discussion

SARS-CoV-2 infection had a major impact on obese people, as the requirement to stay home during the lockdown period had severe effects on health behaviors and well-being in our sample of individuals with obesity [27,28]. The results demonstrate that COVID-19 is impacting the health of individuals with obesity regardless of whether they contract the infection or not. In particular, the statistical differences in lifestyle questions indicated that the COVID-19 pandemic corresponded to more obesity. 

Most of the individuals with obesity on the waiting list have experienced weight gain, due to a worsening of mood and a negative change in eating habits with an increased intake of unhealthy food. 

It emerged that the long period of lockdown had a negative impact on the nutrition of the patients, who, due to the excessive time spent at home and the deterioration of social interactions, frequently indulged in so-called food “snags”, unhealthy behaviors (abolition of sport activities), and reported a general worsening of anxious symptoms. Nonetheless, the mandatory isolation likely allowed insights on their pathological conditions, leading to a notable increase in their motivation to undergo the bariatric procedure. 

Having said that, as shown from the answers, a good number of individuals increased their daily meals, very often choosing types of suboptimal foods from a nutritional standpoint, justified by confinement and increased stress, due to the this now well-known historical period [13,15]. Literature data report that changes in diet, sleep, and physical activity appear to be related to bad mood and stress caused by the lockdown. Let us say that the various components are connected to each other in a sort of vicious circle; as stress increases, sleep quality decreases, the employment of an unbalanced diet increases, and physical activity decreases [14].

Moreover, one of the most worrying data is precisely the important percentage of obese patients who contracted the infection and had to resort to hospitalization. This leads us to think that the state of the host is fundamental for the outcome of the disease. The chronic state of inflammation of the adipose tissue of the obese subject translates into a greater risk of contracting COVID-19 and a worse prognosis [12,13,14].

In addition, Pellegrini et al. administered a survey to all the patients of their unit with a 12-question multiple-choice questionnaire relative to weight changes, working activity, exercise, dietary habits, and conditions potentially impacting nutritional choices, and it was found out that the adverse mental burden linked to the COVID-19 pandemic might be associated with their increased weight [28]. Sideli et al. demonstrated that most individuals with eating disorders and obesity reported symptomatic worsening during the lockdown [29,30].

Robinson et al. examined weight-related behaviors and weight management barriers among UK adults during the COVID-19 social lockdown by completing an online survey that included measures related to physical activity, diet quality, excess food, and how mental/physical health was affected by the lockdown. Participants also reported perceived changes in weight-related behaviors and whether they had experienced weight management barriers, compared to prior to the lockdown [31].

In Waledziak’s study, it is showed that maintaining physical activity and contact with bariatric care specialists are important factors in avoiding weight gain in patients awaiting bariatric surgery [32].

Shujuan Yang et al. studied changes in obesity and activity patterns among youth in China during the COVID-19 blockade using the COVID-19 impact on lifestyle change survey. They observed that the average body mass index of all young participants increased significantly. Their activity patterns had also changed significantly, including the decrease in the frequency of engaging in active transportation, moderate/vigorous intensity housework, moderate/vigorous leisure time physical activity and leisure walking, and increased sedentary lifestyle, sleep, and screen time [33].

Sidor et al. administered an online cross-sectional survey of adult Poles that showed an increased tendency to eat and snack, and these trends were more frequent in overweight and obese individuals, with weight gain, increased consumption of meat, dairy, and fast food and an increase in the consumption of alcohol and smoking during quarantine [34].

In this scenario, as reported by literature, telemedicine appears to be equally effective to in-person treatment programs, as measured by weight loss, with high rates of patient satisfaction reported [35]. While there have been fewer articles published about the use of telemedicine to treat obesity in adults, it has been suggested as a novel and powerful way to deliver comprehensive patient centered care, and as a tool to increase access to bariatric surgery, where less than 1% of eligible patients undergo this procedure, and half of those who initiate a treatment program drop out [36,37]. 

These observations point to the critical need for implementation of preventative measures during periods of lockdown, particularly when the duration of a lockdown is uncertain [38,39]. Such measures might include implementation of telemedicine lifestyle programs, practitioners of medicine can offer supplemental guidance encouraging families to maintain healthy lifestyle choices, and facilities can be designed for implementing exercise programs that minimize viral transmission [40,41].

While our report sheds light on how pandemic-related restrictions affect health habits and weight, and what can be done about it, there are limitations that need to be considered [42,43]. Our cross-sectional study is based on the survey results and, therefore, should be interpreted with caution and be treated as estimates. Self-reported responses can be influenced by various biases [44].

Taken together, these limitations suggest the need for further study, which is feasible, given that at the time of writing, the restrictions remain in place in many countries around the world [45,46].

## 5. Conclusions

The results of this study give the perception of how the COVID-19 pandemic could have had a significant impact on health parameters, including quality of life, mental health, physical activity, and weight maintenance in individuals with obesity. This highlights how the pandemic has changed health habits, and in particular, it has increased the risk of developing anxiety and weight gain in those who are already obese. Authorities should consider the effects of stopping bariatric surgery, that this may have negatively affected the psycho-physical well-being of obese people. Further studies on this topic are needed to confirm these preliminary results that were obtained from a limited sample size.

## Figures and Tables

**Table 1 healthcare-10-01807-t001:** Statements from patients with obesity on the waiting list for bariatric surgery on the impact of the COVID-19 lockdown on their mental health.

	n (%)	n (%)			Chi^2^	*p* Value
Question 1: Sex	Male	Female				
	37 (71%)	15 (29%)			17.89	*p* < 0.05
Question 2: Occupation	Employed	Unemployed	Housewife	Student		
	13 (25%)	20 (39%)	11 (20%)	8 (16%)	51.69	*p* > 0.05
Question 3: Age	21–30 y	31–40 y	41–50 y	51–60 y		
	13 (25%)	17 (33%)	13 (25%)	9 (17%)	60.28	*p* > 0.05
Question 4: Has the obligation to stay at home due to the pandemic affected your mood in a negative way?	yes	no				
	30 (58%)	22 (42%)			2.56	*p* > 0.05
Question 5: Has the obligation to stay indoors due to the pandemic affected your eating habits?	yes	no				
	31 (60%)	21 (40%)			4	*p* < 0.05
Question 6: During the lockdown in the spring, spending all day indoors, were you prone to take in more unhealthy foods?	yes	no				
	33 (63%)	19 (37%)			6.6	*p* < 0.05
Question 7: During the lockdown, did you spend more time in the kitchen at the stove to spend your time?	yes	no				
	23 (44%)	29 (56%)			1.44	*p* > 0.05
Question 8: Do you think that the ban on playing sports outdoors and in company and the closure of gyms has worsened your obesity condition?	yes	no				
	37 (71%)	15 (29%)			17.64	*p* < 0.05
Question 9: Would it have been helpful to have online interviews with the team psychologist and nutritionist?	yes	no				
	29 (56%)	23 (44%)			1.44	*p* > 0.05
Question 10: Did you contract SARS-CoV-2 virus infection while awaiting surgery?	yes	no				
	3 (6%)	49 (94%)			77.44	*p* < 0.05
Question 11: If you contracted SARS-CoV-2 infection, did you need hospitalization?	yes	no	not contracted			
	0 (0%)	3 (6%)	49 (94%)		94.18	*p* < 0.05
Question 12: Did waiting for bariatric surgery during the lockdown lead you to be even more convinced to want to operate or did it demotivate you?	unmotivated	convinced				
	0 (0%)	52 (100%)			100	*p* < 0.05
Question 13: Did a state of anxiety arise during the lockdown period?	yes	no				
	36 (69%)	16 (31%)			14.44	*p* < 0.05
Question 14: Did your cohabitants during the pandemic help you in trying to limit yourself in intaking an excessive amount of calories or did they advise you to indulge in food to reduce anxieties and fears?	they helped	they didn’t help				
	29 (56%)	23 (44%)			1.44	*p* > 0.05
Question 15: Do you think it was right to stop bariatric surgery during the COVID-19 period?	yes	no				
	36 (69%)	16 (31%)			14.44	*p* < 0.05
Question 16: Did you experience weight gain during the lockdown?	yes	no				
	41 (79%)	11 (21%)			33.64	*p* < 0.05

## Data Availability

Data is contained within the article. Authors can use this data for research purposes only by citing our research article.

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
