# Peer review of "Lifestyle in Obese Individuals during the COVID-19 Pandemic"

_healthcare, 2022, doi:10.3390/healthcare10091807_

Round 1

Reviewer 1 Report (Previous Reviewer 1)

The authors made some changes proposed by the reviewer; however, I consider that the article still does not have the conditions to be accepted for the following reasons:

- I think the study’s title and objective should be reformulated based on the methodological design described in the manuscript. In fact, it is a descriptive study that sought to know the behavior (diet, physical activity), needs or daily routines of subjects with obesity on the waiting list for bariatric surgery in the Department of Medical and Surgical Sciences during the Covid-19 pandemic.  To assess the impact, the authors should have a more robust methodological design with a comparative group (for example, before or after covid confinement).

- the description of the construction and application of the questionnaire from a methodological point of view should be improved in the manuscript (was there any pilot study? Is there evidence of test-retest reliability?

- Do you consider that the questions in the questionnaire assess physical and mental health or just behaviors? We know that the concept of physical and mental health is much more complex (multidimensional).

- in summary, I think that the study’s objective and methodology should be reformulated, which will only undergo a descriptive study (analyze/describe the behavior of lifestyles of subjects with obesity on the waiting list for bariatric surgery in the Department of Medical and Surgical Sciences during Covid-19 pandemic.)

Author Response

Thank you for your support.

In attach, our response. 

Reviewer 2 Report (New Reviewer)

Introduction

1.     The text should be reviewed in terms of spelling and grammar. eg. The dots in front of the parenthesis below should be deleted.

impact on lifestyle. [5,6].

population. [17].

2.     For example, the use of “%” in the text differs.

25% were employed, 20 % were housewive

3.     In the paragraph, information about the effect of the Covid-19 virus on the lungs in general is given. This paragraph is not primarily relevant to the subject of the study. This section can be simplified or removed.

4.     At least 2 references should be given for sentences such as: (örn. several recent studies)

Several recent studies have 51 reported that COVID-19 caused the destruction of lung parenchyma, including inter- 52 stitial inflammation and extensive consolidation [9].

5.     It is recommended to use "individuals" instead of "subjects".

Methods

6.     Why weren't all the patients on the waiting list included in the study? This should be clearly stated in the text.

7.     In English texts, dots should be used instead of commas in numerical values. All text must be edited. E.g. (p<0,05)

Discussion

8.     How was the following conclusion reached?

“Most of the subjects with obesity on the waiting list have experienced weight gain, due to a worsening of mood and a negative change in eating habits with an increased intake of ultra-processed food”.

9.     Sentence patterns such as the following should not be used in the discussion section. “The p value, extrapolated from chi-square test, reported statistical significance related to the question number 1-5-6-8-10-11-12-13-15-16, as shown in table 1”.

10.  The results of the study have not been adequately discussed with the literature and the entire discussion section needs to be reviewed.

Author Response

Thank you for support.

In attach our response. 

Reviewer 3 Report (New Reviewer)

Dear Editor,

I have read with great interest the manuscript proposed by Pavone et al., entitled: "How do COVID-19 stay orders affect the physical and mental health among obese patients?". However, there are some issues taht need to be addressed before further processing.

Introduction:

- Lines 67-68: I suggest reporting data on obesity and mental health, as reported by this recent other MDPI paper:  https://doi.org/10.3390/jpm12071106

- Lines 84-94: I suggest also reporting data on COVID and mental health in general and frail populations: https://doi.org/10.3390/healthcare9070893

Materials and Methods

- Lines 119-121: please cite the SICOB guidelines if mentioned in the text.

- Lines 123-133: how were the question selected? Based on what criteria aim? The whole team proposed them?

- Lines 136-141: is p-value two-tailed?

Author Response

Thank you for support. 

In attach our response. 

Round 2

Reviewer 1 Report (Previous Reviewer 1)

Despite acknowledging the authors' efforts to improve the manuscript, I remain disappointed in several respects:

- Some aspects of formatting and spelling errors still need to be improved (see notes in the manuscript)

- In the methodology section, the questionnaire is still poorly described (previous validation? pilot application?)

- the authors changed the title and objective of the study, which focuses on describing the lifestyle behaviours of obese individuals during confinement, based on a questionnaire, i.e., based on patients' perceptions.

This change pleased me. However, the authors should be more rigorous in describing and interpreting the results/discussion. The statistically significant differences found must be analysed correctly. For example, the ones on lines 171 -175 refer to the following:

“Most of the individuals with obesity on the waiting list have experienced weight gain, due to a worsening of mood and a negative change in eating habits with an increased intake of ultra-processed food. We have extrapolated these observations from the results obtained from the questionnaire which show statistical significance for the answers number 1-5-6-8-10-11-12-13-15-16”

In my opinion, questions 1, 10 and 11 do not support what the authors referred.

- Table 1 remains poorly structured. Questions 1, 2 and 3 should not be included in the table, as they are data that are included in the description of the study sample.

- the conclusion should be more modest, as the authors did not assess the impact, but only patients' perceptions of their lifestyle during confinement.

Author Response

Thank you for your suggestions. Please see the attached file 

Reviewer 2 Report (New Reviewer)

.

Author Response

We thank the reviewer for your feedback. 

This manuscript is a resubmission of an earlier submission. The following is a list of the peer review reports and author responses from that submission.

Round 1

Reviewer 1 Report

The manuscript presents itself with quality to be accepted in the Healthcare journal. Since its first version, it has undergone a substantial improvement in its formatting and content (introduction/methodology/results and discussion).

In my opinion, the manuscript can be accepted after improving the formatting of table 1. I suggested that the table have fewer lines and that the results are presented as follows for each question: n (%)

Throughout the text, I noticed that the p (significance level) is presented either capital or lower case. Authors must put all p in lowercase.

Author Response

thank you 

Reviewer 2 Report

I read this paper with great interest. I believe it is vital to study potential side effects of government-imposed lockdown on mental and physical well-being. That said, the current study suffers from several limitations which impede its publication.

At a theoretical level, it is unclear why the study focuses on obese people, and why it is important to examine other health-related effects of the lockdown within this group. I encourage the authors to search for a overarching theoretical framework which could form the basis of their investigation.

Second, the article needs careful proofreading. I found numerous linguistic errors, I copy-pasted a few examples below, but please also check the remainder of the manuscript:

-        “Regarding COVID-19 disease, the subjects with obesity are more expose to risk 69 of development this disease.” à something is wrong with this sentence grammatically.

-        The same goes for the next sentence (where ‘as’ should be ‘is’, I think).

-        “It is knowing the possible link between obesity and COVID-19 and its impact on 79 molecular mechanism.” à again, this is a very strange sentence.

-        “and they completed the questionnaire (86,7%)” à does this mean that 86.7% responded to the survey? Please be specific in the information you share with the reader.

-        “There were 17 (33%) of people were aged between 31 and 40 years” à please rewrite in correct English.

-        “Among of patients the 58% stated that the pandemic negatively affected their mood” à again, please rephrase correctly.

-         

Statistically speaking, please be consistent in the description of numerical info. Sometimes, percentages are rounded (e.g., 33%), other times they are written with two numbers after the comma, then again with only one…

Fourth, and perhaps most importantly, there are a number of methodological problems with this work. To begin, there is very little information about the selection of the items. How was this done? Were there any previous academic papers that informed the selection process? Who created the items and were they pretested in some way?

I realize it is hard to find a large sample of respondents from a particularly vulnerable group and to follow these longitudinally, but did the authors consider recruiting a control group? I wonder if the same responses would be found among the general population and what this would mean. In other words, it comes back to the core question: What was the exact purpose of this study? In its current form, it just shows that for some items, a percentage significantly different from 50% indicated that they had this experience, whereas for other items it was approximately fifty-fifty. How does this inform us scholars and practitioners? And since we are not sure this is a trend specifically found in this target population, what can we do with these findings?

Finally, in the discussion section, the authors mention that being obese increase the risk of being hospitalized when infected with COVID. Yet, none of the respondents in their sample were hospitalized. How do the authors reconcile this? What could be the reason for this counterintuitive result in their study?

Author Response

thank you 

Round 2

Reviewer 2 Report

None of my suggestions have been adequately incorporated and all major limitations of the study remain. Hence, I believe this paper is not suitable for publication.